# Security Ontology Structure for Formalization of Security Document Knowledge

**Simona Ramanauskaitė** [1,*] , **Anatoly Shein** [2], **Antanas Čenys** [3] **and Justinas Rastenis** [3]

1 Department of Information Technologies, Vilnius Gediminas Technical University, LT-10223 Vilnius, Lithuania
2 Orchestra Group, Tel Aviv-Yafo 6688314, Israel; shein@orchestra.group
3 Department of Information Systems, Vilnius Gediminas Technical University, LT-10223 Vilnius, Lithuania; antanas.cenys@vilniustech.lt (A.Č.); justinas.rastenis@vilniustech.lt (J.R.)
* Correspondence: simona.ramanauskaite@vilniustech.lt

**Abstract:** Cybersecurity solutions are highly based on data analysis. Currently, it is not enough to make an automated decision; it also has to be explainable. The decision-making logic traceability should be provided in addition to justification by referencing different data sources and evidence. However, the existing security ontologies, used for the implementation of expert systems and serving as a knowledge base, lack interconnectivity between different data sources and computer-readable linking to the data source. Therefore, this paper aims to increase the possibilities of ontology-based cyber intelligence solutions, by presenting a security ontology structure for data storage to the ontology from different text-based data sources, supporting the knowledge traceability and relationship estimation between different security documents. The proposed ontology structure is tested by storing data of three text-based data sources, and its application possibilities are provided. The study shows that the structure is adaptable for different text data sources and provides an additional value related to security area extension.

**Keywords:** security; ontology; structure; formalization

## 1. Introduction

The development of modern information and communications technologies (ICTs) brings new possibilities for users and organizations, whereby the user is not strictly attached to physical data storage, can access their data anytime and anywhere, use different methods and services for data processing and sharing instantly, etc. Together with the ITC possibilities, the variety of cyberattack vectors has also increased. This is expected because of the complexity of modern technologies, as well as orientation to user experience (UX). Therefore, the spending on security and risk management increases every year, reaching 155 billion USD worldwide in 2021 [1].

The growth of spending on security and risk management is affected by multiple factors [2]: transition to remote or mixed working; cloud, SaaS security assurance; the rise of new threat landscapes. A solution to fight the current spending needs on security and risk management is cyber intelligence. In cyber intelligence, artificial intelligence (AI) solutions are used to automate the process, while providing additional benefits to specific security and risk management areas [3,4].

The development of cyber intelligence is affected by a lack of data for data analysis and decision support. While supervised learning AI solutions are mostly oriented on some specific tasks (data classification, anomaly detection), ontologies as a knowledge base for process automation might have a wider application (semantic modeling, extraction of needed knowledge, etc.) [5].

The ontology structure defines the simplicity of knowledge extraction, while the real value of the ontology relies upon the data it stores. The biggest portion of security knowledge at the moment is not structured; it is presented as text data and is, therefore,

currently limited for application in cyber intelligence solutions. It is important to have a mechanism, assuring a wide range of up-to-date and qualitative data from different sources it. Manually updating security ontology is not practical because of the wide variety of data sources, potential impact of data interpretation, lack of resources, etc. Some methods for text transformation to ontology exist [6]; however, they concentrate on the estimation of concepts, instances, hypernyms, and hyponyms, with no relationship between the data source and concept. When adopting ontology knowledge application and decision justification by mapping knowledge to appropriate data sources, the ontology structure has to be suitably designed.

This paper aims to increase the possibilities of ontology-based cyber intelligence solutions by presenting a security ontology structure for data storage to the ontology from different text-based data sources, supporting the knowledge traceability and relationship estimation between different security documents. Therefore, the main contribution of the paper is answering the research question regarding the main principles of text-based security document formalization to the ontology for gathered data usability and generation of new knowledge.

The paper reviews related works on security ontology and text transformation to ontologies. On the basis of the review results, a new security ontology structure is proposed to provide a linking of the concepts to original data sources. The proposed structure is validated by presenting some numerical results of its application and directions of usage of such an ontology structure.

## 2. Related Works

"An ontology is a formal and explicit specification of a shared conceptualization" [7]. It is a basis of semantic modeling and allows the storage of different concepts, as well as their properties and relationships. Therefore, ontologies are known as knowledge bases rather than databases. Because of the properties of ontologies, they represent one of the solutions for cyber intelligence and a future research direction [8]. The potential of ontologies can be seen in different application areas, such as digital evidence review [9], software requirement and security issue detection [10], modeling of Internet of things design [11], security alert management [12], and as a standard for cyber threat sharing [13].

Ontologies are mostly created by area experts. The expert designs the ontology by formalizing its knowledge using different data sources. Ontologies based only on expert knowledge mostly present the landscape of an area, while additional tools and transformations are used to incorporate existing knowledge into the structure of the designed ontology. Ontologies, with formalized knowledge of different sources, have a higher value, as they present not only the general concepts of the area but consolidate knowledge of different data sources and serve as a knowledge base. However, the transformation from different data sources to ontology might be complicated because of different data formats and types. One of the most complex data types for formalization is text-based data. The same knowledge can be presented in very different texts, and word-to-word matching might not be enough for knowledge matching. Therefore, it is important to find the best solution for text-written knowledge extraction and transformation to ontology.

The next two sections are dedicated to analyzing the existence of security ontology, as well as presenting knowledge of different security area documents and existing solutions to transform text-written knowledge to ontology.

### 2.1. Security-Related Ontologies

The number of publications on security ontology-related topics in the Web of Science Core Collection has increased every year. Analyzing the publication number in the period between 2000–2021, the distribution of publications containing the terms "cyber ontology" or "security ontology" and publications containing the term "security" was very similar year by year (see Figure 1 below). Despite the number of publications on the general term "security" being more than 400 times higher (572,311 records for "security", compared to

2663 records for "cyber ontology" or "security ontology"), the tendencies were the same, i.e., the popularity of the topic is increasing in scientific publications. This indicates that the security ontology topic has been analyzed in scientific papers with the same growth as security in general.

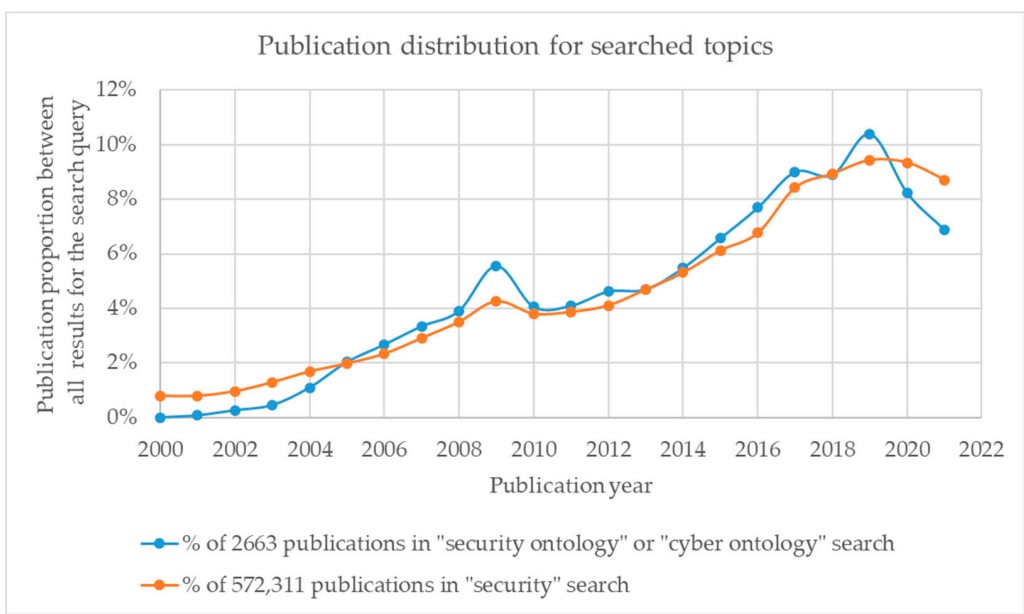

**Figure 1.** Growth of security ontology and security-related scientific papers.

Some of the analyzed papers proposed a new security ontology, others presented solutions based on the ontology or simply reviewed the current situation in the landscape of existing ontologies. One of the first attempts to present a general-purpose security ontology was by Schumacher [14]. The author presented nine concepts and relations between them. The same basic structure of concepts was applied by Tsoumas and Gritzalis [15] to present an idea of security management, based on security ontology. Since this time, the variety of security ontologies has increased and been directed to more specific areas of security, such as for annotating resources [16], for corporate assets and their threats [17], for incident analysis [18], for security requirement elicitation [19], for cloud security [20], for Internet of things security [21], and for the cross-site scripting attack [22].

One of the ways to increase the content of the ontology is to incorporate the data and content of some existing security-related systems. Example of such data sources are the CVE (Common Vulnerabilities and Exposures), CWE (Common Weakness Enumeration), CPE (Common Platform Enumeration), and CAPEC (Common Attack Pattern Enumeration and Classification) [23,24]. These sources have a clear structure and discrete values for specified attributes. Therefore, the transformation of the data to security ontology does not require intelligent solutions.

A big portion of security knowledge is presented as text in security standards and best practices. These security-related documents are also incorporated into security ontologies. One of the first cases to reflect security document data was presented by Parkin et al. [25]. These authors incorporated the ISO27002 standard structure (chapter, section, guideline, guideline step) to the ontology, by mapping it to the asset. Several other ontologies were also based on the ISO27002 standard [26,27]; however, all the ontologies were based on manual human work, where the security standard is analyzed, interpreted, and presented in the ontology by a human expert. Furthermore, in most cases, the requirements or guidelines of the security standards were not expressed in very basic and general concepts; they had a higher level of detail and were, thus, not fully adapted for fully automated content extraction. Therefore, solutions for text-based document analysis and transformation to ontology are needed.

## 2.2. Text Transformations to Ontology

Manually designing the ontology is not an option when wide and complex domains are presented and automated tools are needed to simplify the task [28]. Meanwhile, possible solutions for the automated ontology construction from text documents are implemented in different ways. Moreno and Perez [29] relied on statistics when multiple data sources were analyzed to extract the most frequent terms and incorporate them into the ontology. The principle of multiple data sources was used to extract the main knowledge in [30]. This approach is limited as it extracts just the terms identified in multiple sources. Therefore, more specific terms can be missed or ignored. At the same time, the detection of synonyms is very important to prevent ignorance or rarer terms and their synonyms. To solve this problem, some reference sources can be used. For example, in the tourism domain, the named entities are extracted as the main knowledge on the tourism domain ontology, mostly including locations, organizations, and persons [31]. Another option is to use natural language processing (NLP) solutions [32]. In most cases, the part of speech (POS) is estimated, where the nouns are identified as key concepts [33]. The concepts are additionally processed by using synonym tables [33]; however, this can be applied to narrow domain areas, as a detailed list of synonyms can be an issue for more complex domains. In such a case, clustering might be used to organize the concepts, find synonyms, and indicate relations between the concepts [34–36].

The ontology construction can be executed on very different levels to define terms, synonyms, concepts, concept hierarchies, relations, or rules [37]. A more detailed (including all levels of concepts) ontology increases its application possibilities, but also increases its construction complexity. Therefore, research on relation estimation between concepts is an important aspect of ontology construction. Semantic patterns can be extracted to identify relations between concepts [38,39], while grammar-based transformation [40] and supervised learning can also be applied [41,42]. For relation extraction, the semantic lexicon, syntactic structure analysis, and dependency analysis are mostly used [43].

Despite the variety of existing solutions for ontology learning from unstructured text, the performance of the transformations lacks accuracy, better results can be achieved when some specific domain is analyzed [44]. The transformation of unstructured text to ontology according to the domain allows adding some specific rules or solutions, enabling a more detailed presentation of the knowledge [39,45].

In cybersecurity, research on knowledge extraction from text exists [46,47]; however, automated ontology building or enrichment is mostly achieved using different data sources rather than unstructured text [48–50]. In the field of security ontology, Gillani and Ko [51,52] proposed a ProMine solution to enhance or maintain ontologies by using text-mining technologies. These solutions are based on the application of an existing ontology and external data sources (for synonym estimation) to indicate additional concepts from unstructured text. These solutions illustrate the need for a reference ontology to which the extracted data will be added. The ontology main structure is needed to define the main rules for the presentation of the extracted terms and relations. However, security ontology structures and transformations, dedicated to transforming security standards and best practices for data construction of security ontology, are still missing.

## 2.3. Summary of Related Work

The analysis of existing security ontologies and data source transformation methods as knowledge bases revealed that most knowledge source generation is based on manual work involving experts (see yellow blocks in Figure 2). Fully automated solutions [51,52] (presented as a green block in Figure 2) are oriented toward the presentation of security concepts without the presentation of the data source at different aggregation levels. Security metadata and aggregated data sources exist; however, full integration between formalized security concepts and data sources is missing in the existing security knowledge data sources (see *X*-axis in Figure 2).

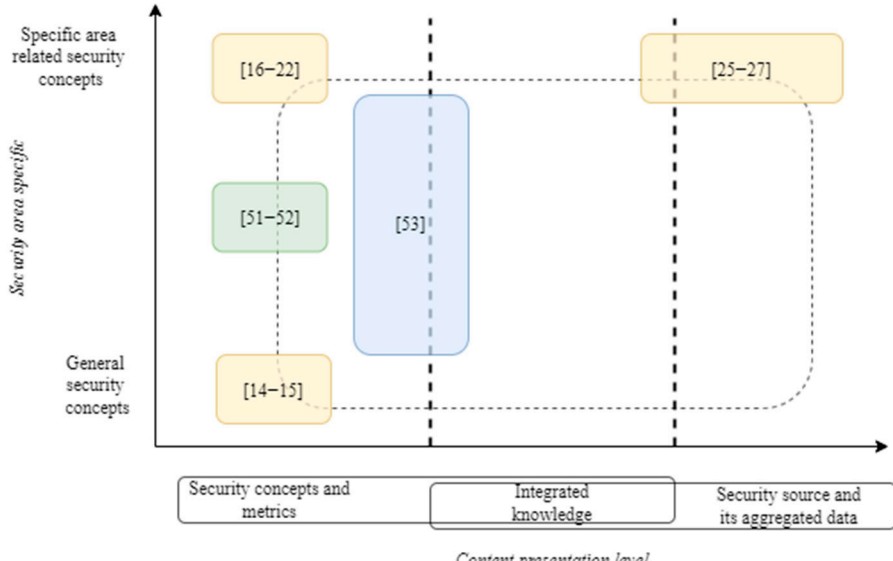

**Figure 2.** Summary of analyzed related words, based on presented security area and concept presentation level.

UCF Mapper [53] has a semi-automatic solution (presented in blue in Figure 2) when initial security text analysis is executed automatically, while human work is used to adjust the knowledge. This solution defines security document controls, linked together by using security concept similarity. The integration between security concepts and security documents exists but is implemented at a very abstract level only.

Further examples of partly integrated knowledge between security concepts and documents are security ontologies for mapping of security standards [25–27]. Those ontologies are mostly oriented toward security documents and aggregated knowledge, with just some links to formalized security concepts. This complicates their usage by automated systems; therefore, to realize the full potential of security document formalization, knowledge of different abstraction levels should be presented with its interconnections.

## 3. Security Ontology Structure for Text-Based Security Source Formalization

A security ontology for text-based security source formalization should define the structure and principles for knowledge extraction from different sources. This would allow automated composition of the security knowledge base using multiple data sources, rather than the perception of the ontology developer. Such a knowledge base presented as an ontology might serve as a base for different security intelligence tasks.

One of the requirements in modern knowledge and decision support systems is a justification of the decision. To do so, some relations between the data source and extracted terms, as well as the associated concepts, should be implemented. At the same time, the data source can give additional value and clarity for the decision traceability. Therefore, the structure of the proposed security ontology has three main layers (see Figure 3): data sources, including structure and content (documents); security concepts, as well as their properties and relations between concepts (knowledge); relations between data source and security knowledge, expressed as atomic sentences with links to concepts (mapping).

To present the security source, the document structure is important. Division into sections, subsection, controls, description, and other components is standard for a well-written security document; therefore, it should also be reflected in security ontology. However, different security documents might have different structural elements. This complicates the alignment of several data sources. Therefore, we ensure the adaptability of the security ontology structure to different data sources by applying class inheritance. The reference structure for data sources is composed of main classes and properties. Figure 4 illustrates the main structure of the ontology, where blue notated elements define main

classes and gray elements denote properties, associated with an appropriate class. Each document is presented as an instance of the "security document" class, while its structure is presented as a hierarchical structure of chapters (presenting a tree of iterative chapters and subchapters). The content of the document should be defined on the basis of the type of content: control, testing procedure, attack description, and definition. Each text element has a "text" property and allows the presentation of the full, not formalized content. For the formalization, each text is divided into atomic sentences, as a link from the document structure to the security concepts.

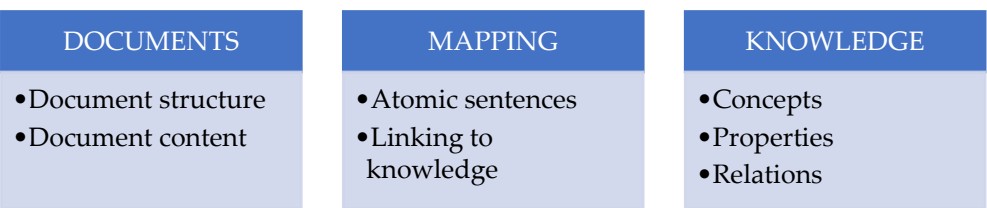

**Figure 3.** The main layers of the proposed ontology structure.

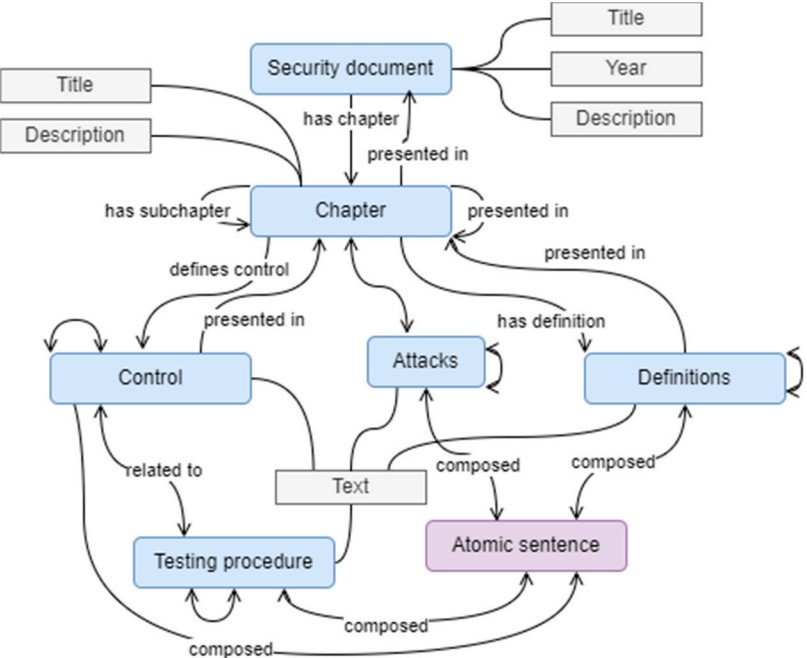

**Figure 4.** The main structure of security ontology data source layer: blue elements—classes; gray elements—data properties of the class; purple elements—class of mapping layer; arrows—data properties, connecting the instances of separate classes.

When a new data source is added to the ontology, new classes should be created for each of the actual (having an analogue component in it) classes in the security ontology data source reference structure. This will allow using reference classes for the selection of data in all inserted data sources. On the other hand, for more specific, defined data sources, the child classes can be used.

Close to the adding of inherited classes, the security document should be presented in the ontology, by creating instances of the classes. Instances reflect the object and data source rather than their structural elements. The naming of the instance can reflect the data source title for a faster search. One instance of security document class is created, in addition to one for each chapter and other elements of the data source.

Classes of the instance define the data source structure and go from abstract instances to more detail, where not only is the title presented but also the text defining the control, testing procedure, or concept description. The text is difficult to analyze as it contains

multiple sentences, whereby one sentence might include different concepts on security. Therefore, each text should be divided into sentences. This can be achieved by using such solutions as finite state machines, part-of-speech tags, conceptual graphs, domain ontology and dependency trees, etc. The main functionality of sentence structure analysis and identification of the subsentence can be completed using widely available programming toolkits, such as NLTK. This simplifies the text division into sentences and later into atomic sentences, to be presented in the mapping layer.

In the mapping layer, each sentence should be divided into atomic sentences. In Figure 5, the sentence is presented in white, while atomic sentences are presented in red. Atomic sentences should present only one idea, without any side sentences. In one sentence, several atomic sentences might be presented and linked by some keywords. The keyword should also be used to link the atomic sentences in the ontology (in Figure 5, the link "leads to" defines the link between two atomic sentences of the same composite sentence). Consequently, appropriate object properties should be used, or an event should be newly added to the ontology if the analyzed sentence has a different keyword than presented in the ontology. However, to assure the ontology data's adaptability to machine usage, keyword processing should be used to eliminate nonmeaningful terms and term conversion to standard form. Therefore, "as a means of" could be converted to the term "leads to" or similar.

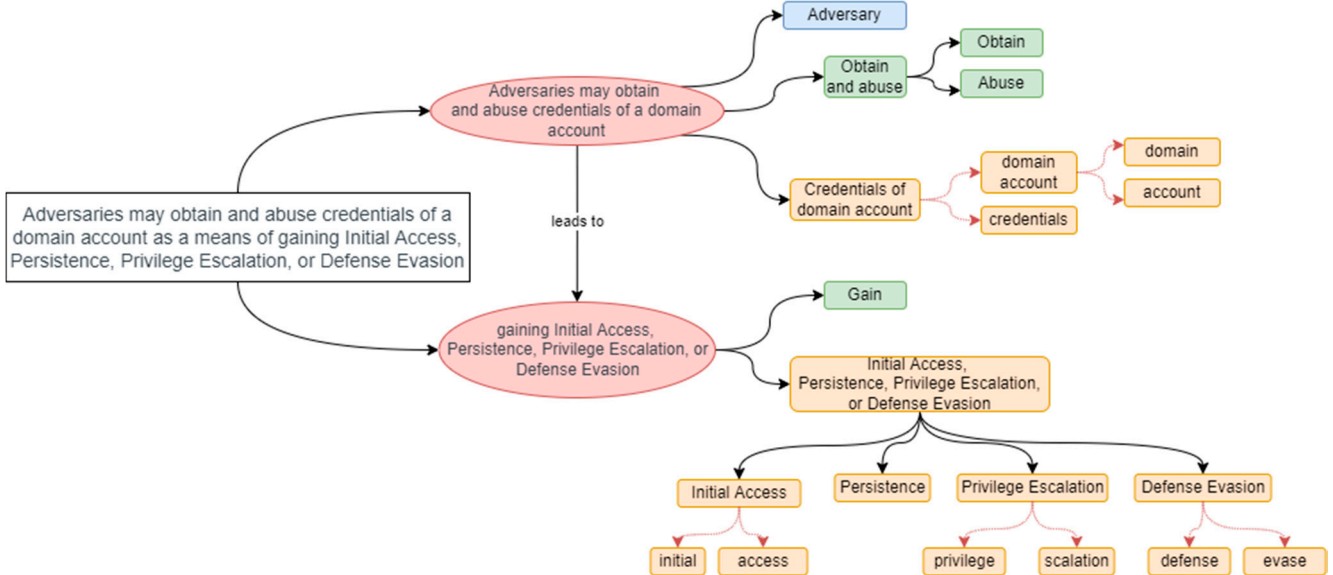

**Figure 5.** An example of sentence presentation as multiple atomic sentences, where each sentence is divided into subject, action, and object, as well as segmented to the lowest-granularity elements.

While atomic sentences present separate ideas and might indicate the logical sequence of concepts, text-based expression is not effective for machine usage. Therefore, each atomic sentence is divided into smaller elements, identifying the subject (blue in Figure 5), action (green in Figure 5), and object (yellow in Figure 5) in it. Such a separation allows an estimation of who is acting, what they are doing, and what object they are using for it. This division might be executed by a human or by natural language processing (NLP). Human-based transformation might be more accurate, as security experts might understand the meaning of the atomic sentence and express some terms in more popular synonyms (for a better match with other data sources). However, this is very time-consuming and, in some cases, requires not general, but very specific security knowledge and situational understanding. Therefore, NLP solutions for subject–object–verb extraction can be adapted for content extraction automation.

To make the content usable for machine systems and concept matching between several data sources, each subject, action, or object is divided into the lowest-level part-of-speech

elements. The hierarchical structure of security concepts is adapted to present the idea, whereby a combination of several concepts might be differently interpreted in comparison to the sum of separate concepts. For example, "firewall and router configuration" might not be identical to the sum of "firewall configuration" and "router configuration", as the interdependencies between those two might also be considered. At the same time, the division into lower-level part-of-speech elements allows an estimation of concept similarity, with not only a full, but also a partial match.

The division of subject, action, and object into smaller elements covers the security knowledge layer. The terms for this layer are added by incorporating new data sources and identifying new, non-existing instances, which are needed to reflect the atomic sentence. At the same time, the object properties are important in this layer. The composite term is divided into lower-granularity terms according to NLP principles. Therefore, the object properties between concepts might indicate the logical operator (and, or, not), property of the elements, etc.

An example of text-based data division into atomic sentences and smaller components is presented in Figure 6. It presents the first two sentences of MITRE ATT&CK technique T1003.001, stored as four atomic sentences (two for each of the sentences). Additionally, the subject of the attack is presented in the purple background to help the identification of relations between system and attack behavior. This situation indicates that the storage of credential material in LSASS process memory is sensitive and directly related to attack actions, thus deserving attention for security assurance. The situation illustrates the formalization principle when two different sentences can be associated by analyzing the linking to the same security concepts. In the same manner, more sentences, chapters, or even security sources can be linked.

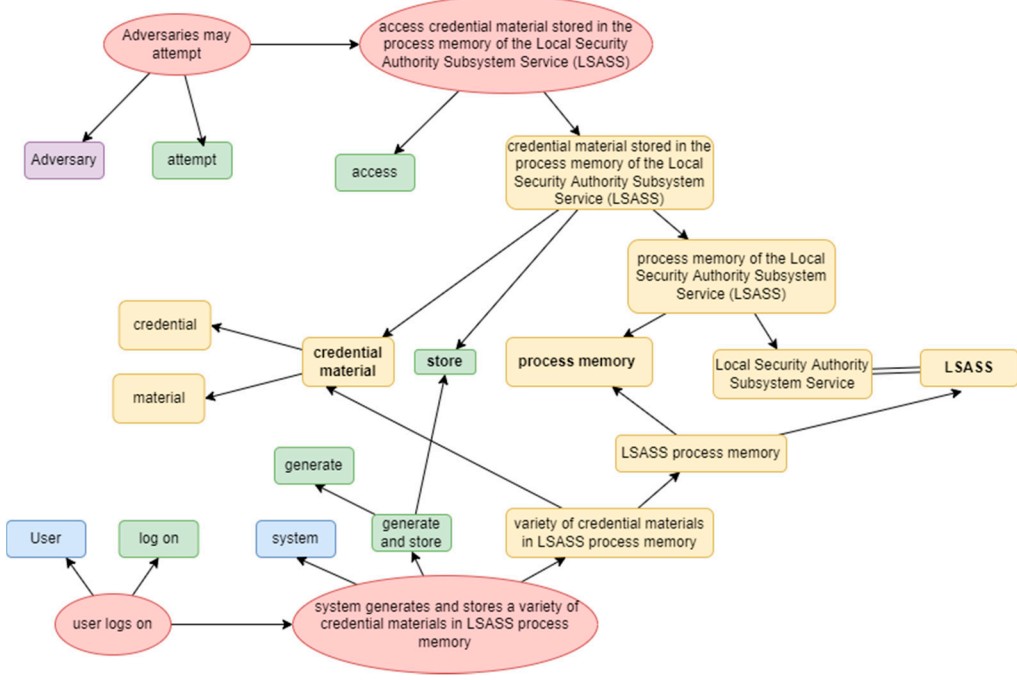

**Figure 6.** An example of attack technique description formalization, by indicating matching concepts between two sentences and the attack relation to system behavior.

The traceability to elements with a higher abstraction level helps identification of the flow, and this can be used for estimation of the distance between different elements. At the same time, such a structure is not optimized in the sense of data storage; it stores full text and duplicates its parts in lower-level elements. This solution is more oriented toward data usability rather than storage optimality. However, to solve the issue, the ontology data can be transformed, by filtering out unnecessary elements, i.e., leaving only the elements

of the needed level (security document structure, atomic sentences, lowest-granularity elements, etc.).

## 4. Application of the Security Ontology and Its Data

This paper presented a security ontology structure and principles of how this ontology should be supported with data from different data sources. It is difficult to compare it to existing ontologies. This paper presented the reference structure for different security document presentations, while other existing security ontologies were mostly dedicated to knowledge presentation. Therefore, several approaches were applied to analyze the security ontology structure suitability to store knowledge of different text-based data sources, as well as its applicability.

### 4.1. Numeric Results of Sample Data Presentation in the Security Ontology

Structure suitability can be estimated by applying it for the formalization of different security data sources. In the current state, human-based sample data from ISO 27,002 (five chapters, three subchapters, and seven controls), PCI DSS (six chapters, four sub-chapters, two general requirements with four detailed requirements, and three testing procedures) standards, and MITRE ATT&CK enterprise techniques (descriptions of two techniques and two sub-techniques with 10 sentences in total) were added to validate the security ontology structure suitability for different text-based security data sources.

The formalization process does not require the adjustment of the reference security ontology structure. However, not all classes were used for instance creation in different data sources; MITRE ATT&CK techniques did not require chapter presentation, while security standard requirements and testing procedures were mostly presented, not attack techniques.

To review the specifics of the mapping layer, the results revealed (see Table 1) 1.88 atomic sentences on average for each analyzed sentence (requirement, testing procedure, control, technique descriptions). This illustrates the complex structure of the texts, presenting several interconnected concepts.

**Table 1.** Summary of the number of instances in sample data of the analyzed security documents.

| Measurement | Value | | |
|---|---|---|---|
| | **ISO 27002** | **PCI DSS** | **MITRE ATT&CK Technique** |
| Number of instances in document structure | 8 | 10 | 0 |
| Number of atomic sentence instances | 13 | 12 | 24 |
| Number of instances of initial terms (subjects, actions, objects) | 30 | 29 | 55 |
| Number of lower granularity term instances | 64 | 55 | 147 |

At the same time, 2.33 lower-granularity term instances on average (not taking into account the match between different security documents) were generated from first-level composite term instances. This does not accurately reflect the situation as the majority of subjects and actions were presented as one term, while objects were mostly presented as complex structures, requiring hierarchical presentation to lower-level granularity term instances.

The experiment illustrates that text-based data (technique descriptions) were written in a more complex manner (usage of complex sentences and terms), but the same term was more often used (probably because the technique description was longer or contained more sentences) in comparison to analyzed security standard requirements and testing procedures.

### 4.2. Analysis of Knowledge Extraction Possibilities Using the Proposed Security Ontology

Suitability to store different text-based security data sources is not enough if there are no use cases of the presented data. Therefore, we present some use cases where the security ontology, with data integrated from different data sources, can add value in comparison to

existing solutions. The list is not limited to these examples; however, it presents the most relevant, easily implementable application use cases.

### 4.2.1. Summary of Data Source Coverage or Security Landscape

To understand what concepts one or several selected data sources cover up, the data sources should be read and summarized. Using the security ontology, a list of mentioned terms can be easily obtained. The list can be reduced by adding requirements to provide only the most popular terms. At the same time, the data can be used to understand which area of the security landscape is covered by the data source in comparison to the full landscape of cybersecurity.

Different data sources analyze different aspects of the security area; therefore, the integration of different security data sources enables a wide security knowledge base. With the help of hierarchical term division into lower-granularity (simple words) terms and knowledge of different data sources, the link between different concepts can be established. Therefore, the term graph can be used as a presentation of a wide range of security areas, thereby forming the full landscape of cybersecurity. Such a data source can be used for learning purposes, as well as security area concept interdependencies analysis.

### 4.2.2. Mapping of Security Documents

Mapping of security documents allows a better understanding of what is common between multiple security documents, as well as their uniqueness and specifics. Some solutions to map different security standards using a reference ontology exist [26]. However, they are based on a very detailed security ontology, and the mapping of the security document to the security ontology must be done by a security expert. An automated solution is used by UCF [53], where text analysis is applied to extract the main terms. The mapping between the security documents is mostly implemented by the proportion of matching terms. However, the solution relies on the relational database rather than the ontology for knowledge storage; therefore, opposite statements such as "password is required" and "password is not required" lead to a high similarity because the proportion between matching terms is high.

Using the proposed security ontology structure and the hierarchical division of complex terms can allow more accurate mapping of security documents. The manual labeling of the most appropriate versions of the term used in UCF Mapper can be replaced by automated matching of terms where the relations are established by incorporating different security documents. Furthermore, the links between different granularity terms will allow the identification of opposite meanings or terms, enabling more accurate mapping of security documents.

### 4.2.3. Cybersecurity Threat Modeling

Security threat modeling tools exist; however, the data for the modeling must be manually transformed from different sources to a specified language or model. Xiong et al. [54] used the MITRE ACC&CK matrix as a data source and transformed it into an enterprise system. This allowed security threat modeling, but a manual expert-based transformation of the knowledge had to be implemented. Using the proposed security document formalization, the subject, action, object, and properties, as well as links between them, can be estimated. This might represent a basis for modeling different security situations. For example, the pre-conditions and post-conditions can be easily identified by analyzing the relations between atomic sentences. This can allow the generation of attack graphs or trees. Subject classification can be used to define attack and mitigation relations for security risk evaluation, while attack subject identification can be used to identify attack flows. This would align with the MITRE ENGENUITY attack flow project [55] as the formalization at different levels would enable flow automated identification based on the relations between concepts, while linking to the data source would allow relationship aggregation to the technique level.

## 5. Discussion and Conclusions

Security-related research and ontology applications are experiencing constant growth. However, the absence of fully functioning semantic web- or text-based security data source formalization solutions limits the exploitation of existing data sources in the cyber intelligence area. This paper goes one step further to solve the problem and provides an ontology structure, dedicated to linking the ontology content with a text-based data source.

The division of the proposed ontology structure into three layers allows a separation of the security area content, security document structure with texts, and mapping between the two. Therefore, the data can be easily filtered to use the security area content only, while additional layers can provide additional values, related to links between different data sources, automated mapping between them, etc.

While the ontology structure is suitable for human-based security document formalization, as shown in Section 4.1, the automated solution should be provided for simplification of text data transformation. The current solutions for text transformation to ontology are ideal for estimating concepts and their relations. For application to this security ontology structure, additional adaptation is needed, as document structure and sentence relation analysis must be incorporated.

**Author Contributions:** Conceptualization, S.R. and A.S.; methodology, S.R. and A.Č.; software, S.R.; validation, S.R., A.S. and J.R.; formal analysis, S.R. and A.Č.; resources, S.R. and A.S.; data curation, S.R.; writing—original draft preparation, S.R.; writing—review and editing, A.S., A.Č. and J.R.; visualization, S.R.; supervision, A.S. All authors have read and agreed to the published version of the manuscript.

**Funding:** This research received no external funding.

**Data Availability Statement:** The data presented in this study are available on request from the corresponding author. The data are not publicly available due to copyright requirements of the text data sources (security standards).

**Conflicts of Interest:** The authors declare no conflict of interest.

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
