# Peer review of "Security Ontology Structure for Formalization of Security Document Knowledge"

_electronics, doi:10.3390/electronics11071103_

Round 1

Reviewer 1 Report

The article further develops the ideas for automated knowledge generation related to cybersecurity management. Synthesis of documents is needed for proactive research of the capabilities of cyber adversaries, investigation of security incidents, creation of documentation for compliance with security standards of the environments of specific organizations. Contributes to increasing the possibilities for threat intelligence by aggregating information from diverse sources. 

In addition, the question answered by this study is not clearly stated. In my perception, the article answers the question of whether there is a way to systematically formalize different sources of information in order to organize in an accessible way the knowledge of cyber security and at the same time generate new knowledge. The text does not make clear the differences and added value compared to existing research. Given the workload of security teams, such technology was extremely useful. I dare say this from the standpoint of my long practice in the field of cybersecurity in creating large cyber security systems. Since this problem area is exactly mine, I see many applications of such an idea. I wrote some of them in the comment in the review form. I believe that the conclusions correspond to the statements about the possibilities of the proposed ontology.  

Out of protocol and only for editors. It seems that the authors have no real idea of the deep problems in the practice of cybersecurity, but at the same time, I think they are on the right track.
  Detail comments:

There is a figure outside the text borders;

Last page needs text editing;

Reviewer 2 Report

This paper proposes a new security ontology structure to provide linking of the concepts to original data sources. The proposed structure was validated by presenting some numerical results of its application and directions of usage of such an ontology structure. This paper can be further improved by addressing the following comments.
1. Explanation of figures 2 and 3 is missing. Authors need to interpret the information in the corresponding figures in context.
2. Please use vector figures (especially fig. 3) to avoid quality loss.
3. The experimental part of Section 4.1 lacks evaluation metrics and reference samples to measure the reliability of the proposed ontology. For example N-gram or Recall etc. In addition, the experimental results are obviously insufficient.
4. Authors should seriously consider the innovative nature of the Security Ontology Structure described in Section 3. What is the meaning of traversing the keywords in the atomic sentence? Is energy consumption considered? Are you considering a more efficient processing method?
5. In general, the authors should reduce the general description and propose a more specific approach. In addition, the amount of data for experimental comparison should be expanded as much as possible. More comparison methods also need to be seriously considered.

Author Response

Please see the attachment,

Reviewer 3 Report

A security ontology is specially constructed to enable TAPIO tool to automatically ingest data from a wide range of data sources. Data sources are usually available in several unstructured sources including text, images, graphs, documents, webpages and others. The authors in this paper tries to suggest a security ontology structure to formalize the security document knowledge. The idea is interesting, however, the paper can be improved by addressing several major concerns such as:

  • The paper seems more like a review paper, not an article. I suggest to change the type of this manuscript to review article since experimental/simulation part is missed (or even might not  applicable)
  • The novelty of this paper is unclear. Since cybersecurity ontology is not a novel topic. The paper lacks to provide summarized contribution of the authors.
  • An overview of security ontology with clarification examples (Use visual examples) is crucial section in this paper, and can be presented after the introduction section. 
  • The correlation of  cyber security ontology with cyber attack trees/security surface analysis. An example can be discussed on this regard to emphasize the role of cyber security ontology in the cybersecurity/attacks analysis.
  • A subsection for the potential cyber security ontology tools that can be used to formalize the security document knowledge/Extraction. 

Round 2

Reviewer 2 Report

This manuscript is much better compared with the last version.

Reviewer 3 Report

The majority of my comments have been addressed by the authors. Thanks, i would accept the paper